# Young Age, Liver Dysfunction, and Neurostimulant Use as Independent Risk Factors for Post-Traumatic Seizures: A Multiracial Single-Center Experience

**DOI:** 10.3390/ijerph20032301

**Published:** 2023-01-28

**Authors:** Nicodemus Edrick Oey, Pei Ting Tan, Shrikant Digambarrao Pande

**Affiliations:** 1Rehabilitation Medicine, SingHealth Residency, Singapore 169608, Singapore; 2Department of Rehabilitation Medicine, Changi General Hospital, Singapore 529889, Singapore

**Keywords:** traumatic brain injury, seizures, neurosurgery, rehabilitation, neurostimulant, neuromodulation

## Abstract

We aimed to determine the potentially modifiable risk factors that are predictive of post-traumatic brain injury seizures in relation to the severity of initial injury, neurosurgical interventions, neurostimulant use, and comorbidities. This retrospective study was conducted on traumatic brain injury (TBI) patients admitted to a single center from March 2008 to October 2017. We recruited 151 patients from a multiracial background with TBI, of which the data from 141 patients were analyzed, as 10 were excluded due to incomplete follow-up records or a past history of seizures. Of the remaining 141 patients, 33 (24.4%) patients developed seizures during long-term follow up post-TBI. Young age, presence of cerebral contusion, Indian race, low Glasgow Coma Scale (GCS) scores on admission, and use of neurostimulant medications were associated with increased risk of seizures. In conclusion, due to increased risk of seizures, younger TBI patients, as well as patients with low GCS on admission, cerebral contusions on brain imaging, and those who received neurostimulants or neurosurgical interventions should be monitored for post-TBI seizures. While it is possible that these findings may be explained by the differing mechanisms of injury in younger vs. older patients, the finding that patients on neurostimulants had an increased risk of seizures will need to be investigated in future studies.

## 1. Introduction

Seizure is a known complication following traumatic brain injury (TBI) and may occur early (within 2 weeks post-injury) or late (beyond the 2-week period). Prophylactic anti-epileptics are useful in reducing early but not late seizures [1,2,3,4]. Various classifications have been used to define the type of traumatic brain injury: blunt injuries lead to focal contusion and hematoma, whereas penetrating injuries can result in damage to the skull and underlying brain tissue. In cases of penetrating injuries, seizure occurrence is thought to be a result of cicatrix formation in the cortex [5]. In non-penetrating or closed head injuries, seizure activity is thought to be due to impairment of blood flow, intraparenchymal hemorrhage, and hemorrhagic contusions, which result in hemoglobin breakdown that adversely affects neuronal function and glycolysis, causing negative alterations of excitatory amino acids [6]. Acute-phase response following TBI is characterized by the activation of astrocytes and microglial cells, which then migrate to the injured area and proliferate [2,7]. Acute inflammation is followed by a persistent chronic stage [7,8,9,10,11]. The combination of early and late inflammation and gliosis is thought to be responsible for epileptic activity [2]. Pericontusional areas have also been suggested as electrically active and may become foci for seizures [6,12], with the majority of patients having evidence of the temporo-parietal regions being the source of seizure activity [1,2,13].

Irrespective of pathophysiology, it is generally accepted that post-TBI seizures are associated with a poorer functional outcome; nevertheless, the risk factors that may predict their long-term occurrence are still debated [14]. We sought to characterize potentially modifiable risk factors that may predispose patients to developing post-TBI seizures. We hypothesized that in addition to the commonly known risk factors such as age, there may be other factors (such as drug use) that are hitherto unknown, which may predict the risk and, if modified, may reduce the chances of patients developing post-traumatic seizures. In this study, we analyzed 141 patients with TBI and confirmed, as per previous publications, that the non-modifiable risk factors of younger age, poor initial GCS scores, and the existence of cerebral contusion on brain imaging are indeed risk factors predictive of post-TBI seizures. In addition, we identified novel and potentially modifiable risk factors such as liver dysfunction, active employment, and the use of neurostimulants as potential risk factors that deserve further study.

## 2. Methods

### 2.1. Patient Selection and Biological Parameters

This is a retrospective cohort study of TBI patients who were admitted to the neuro-rehabilitation facility at Changi General Hospital, Singapore, from June 2008 to October 2017. All patients, regardless of racial background (Chinese, Malay, Indian, or other), admitted with a diagnosis of TBI had their neurological status and initial GCS assessed. All patients uniformly received urgent CT brain scans, baseline full blood counts, biochemistry, liver function tests, and coagulation profiles at baseline presentation. Based on the initial and subsequent neurological status and neuroimaging by CT of the brain, appropriate referrals to the neurosurgical team to perform any surgical intervention were made. All the TBI patients were eventually referred to inpatient rehabilitation services, where a multidisciplinary team reviewed the patients throughout the inpatient period and during their subsequent outpatient follow-up, which included an outpatient rehabilitation program, neuropsychological review, and return-to-work assessments. The median follow-up period was 59 months (range 28–86 months; Table 1). All the patients included in the current study were discharged from the rehabilitation facility and were regularly followed up as outpatients for a minimum of 6 months. The subsequent records of hospital admissions, changes in their general physical and neurological status, and changes in treatment regimens were available both electronically and in paper format for all patients.

### 2.2. Inclusion and Exclusion Criteria

Inclusion criteria were: age above 21 years, established diagnosis of TBI, and complete follow-up records. Exclusion criteria were: incomplete follow-up records, patients who were repatriated to other countries, patients less than 21 years of age, and patients with no clear history of TBI.

### 2.3. Ethics Approval

The Singhealth Centralized Institutional Review Board approved this study (2015/3112). Informed consent from the patients was waived due to the retrospective nature of the study.

### 2.4. Data Collection

Electronic and paper medical records of the patients from the time TBI was diagnosed were assessed, and follow-up visits including additional admissions were reviewed. The data collected included demographic details, mechanism of injury, occupation, admission GCS, nature and severity of injury, admission, and subsequent computed tomography (CT) or magnetic resonance imaging (MRI) brain scan findings. Other parameters reviewed were admission electrolytes, coagulation profiles, premorbid medications, and comorbidities.

We also documented the treatment modalities, including medical treatments used for raised intracranial pressure; types of neurosurgical interventions, including intracranial pressure monitor (ICP), burr hole drainage, craniotomy, craniectomy, and/or ventriculoperitoneal shunting, if any. The CT brain scan findings were categorized as cerebral contusion, extradural or subdural hematoma (EDH and SDH, respectively), skull fractures, diffuse axonal injury (DAI), and subarachnoid (SAH) and/or intraventricular hemorrhage (IVH).

During the inpatient and follow-up periods, all medications were reviewed, with the use of neurostimulation with levodopa or bromocriptine and the use of lipid lowering agents being particularly documented. Liver function tests were defined as abnormal if the patient’s alanine aminotransferase (ALT) and aspartate aminotransferase (AST) levels were higher than two times the upper limit of normal (ULN).

Primary prophylaxis for early seizure was used in all patients for 2 weeks post-initial injury, and any seizure occurring beyond a 2-week period post-initial injury was documented. Seizure was clinically diagnosed either by a physician, neurologist, neurosurgeon, or rehabilitation physician with or without EEG. Treatment initiated and type of anti-epileptic medications used were documented.

### 2.5. Statistical Analysis

Categorical data are presented as frequency (percentage), and continuous data are presented as mean (± standard deviation) for parametric distributions and median (± interquartile range) for non-parametric distributions. The differences in characteristics were examined using chi-square tests for categorical variables and the two-sample *t*-test or Mann–Whitney U-tests for continuous variables, where appropriate. A two-tailed *p*-value of < 0.05 was considered to be statistically significant. The analysis was performed using the Statistical Package for the Social Sciences (SPSS) version 19.0 (IBM Corp. Armonk, NY, USA).

## 3. Results

### Descriptive Analysis

We studied 151 patients with TBI, of which 10 were excluded (8 were lost to follow-up and 2 due to known past medical history of seizure). Of the remaining 141 patients, 24.4% (33) had seizure occurrence during the follow-up period (median = 59 months). The demographic details and admission characteristics are described in Table 1. In the univariate analysis, statistically significant associations with seizures were the presence of cerebral contusion (*p* = 0.031), Indian ethnicity (*p* = 0.051), employment (*p* = 0.028), abnormal liver function on admission (*p* = 0.004), neurosurgical intervention (*p* = 0.085), low GCS on admission (*p* = 0.001), prolonged hospitalization (*p* = 0.005), and use of neurostimulant medications (*p* = 0.001). Multivariate analysis revealed that patients of younger age (OR = 0.96, 95% CI: 0.93–0.98, *p* = 0.001) and those who used neurostimulants (OR = 10.9, 95% CI: 3.5–34.2, *p* ≤ 0.001) were at increased risk of post-TBI seizure occurrence.

Multivariate logistic regression was used to predict the factors contributing to seizure. The forward selection method was applied for the following factors: age, race, presence of liver dysfunction, admission GCS scores, and use of neurostimulants.

The final model revealed that younger age (OR = 0.96, 95% CI: 0.93–0.98, *p* = 0.001) and those who used neurostimulants (OR = 10.9, 95% CI: 3.5–34.2, *p* ≤ 0.001) were at increased risk of seizure occurrence.

## 4. Discussion

### 4.1. Summary and Contributions

Amongst the 141 patients with traumatic brain injuries analyzed in this study, seizures were noted in 33 (24.4%) patients, despite receiving primary prophylaxis for 2 weeks following initial TBI. This figure is slightly higher than previously reported incidences of early post-TBI seizures [6,15,16]. Multivariate analysis shows that age, neurostimulant use, and liver dysfunction all had significant effects on predicting whether a patient would develop post-TBI seizures (Table 2). 

In terms of injury pathophysiology, patients with cerebral contusions were found to have a higher level of seizure occurrence (22 vs. 51; *p*-value = 0.05). In contrast with previous studies that found a higher incidence of seizures following TBI in patients with ICH and SAH [6], we did not find any significant correlations of post-TBI seizures with other types of intracranial hemorrhage such as SAH or IVH (Table 1). One possible confounder is that we might not have detected nonconvulsive seizures, which reportedly account for 50% of seizures in intensive care units following TBI and which may lead to raised ICP beyond 96 h post-initial injury. As seizures may lead to metabolic disturbances causing cellular injury, continuous EEG monitoring would be useful to monitor and maintain ICP within a safe range; prompt control of seizures as they occur would be extremely important, as past studies suggested that status epilepticus increases mortality [6]. Due to the retrospective nature of our study, we did not have accurate information on seizure occurrence during the immediate post-brain-injury, intensive care unit, or peri-operative periods, and we were therefore unable to analyze the relationship of early vs. late post-TBI seizures.

In terms of medical risk factors for post-TBI seizures, a Finnish study previously found an association between late post-traumatic seizure with depressed skull fractures and the presence of early seizures [14]. The authors did not find any association of seizures with admission GCS, duration of unconsciousness, or duration of post-traumatic amnesia (PTA). In contrast, our data suggest that low GCS at the time of admission was associated with post-traumatic seizures (mean GCS of 9.5 compared with 11.9 in the no-seizure group; *p*-value = 0.003). We also noted that a longer length of stay was associated with post-TBI seizure occurrence (mean of 53.1 days versus 36.3 days in the no seizure group; *p*-value = 0.021). The most probable explanation for this finding may be that those patients with more severe brain injury needed neurosurgical interventions (37.5% vs. 51.9% in our data set; *p*-value < 0.001), which may also be associated with perioperative complications and hence longer lengths of stay.

### 4.2. Strength and Limitations

With respect to other factors influencing the risk of post-TBI seizures, studies have identified that the mechanism of injury matters: increased seizure frequency is associated with penetrating brain injuries [17]. From a long-term follow-up study over a period of 50 years, one group found that the risk of seizures increased with the severity of initial brain injury, as well as the presence of SDH, skull fracture, PTA duration more than 24 h, and age above 65 years [18]. In terms of imaging, hippocampal atrophy were observed to be significantly associated with seizures [19]. Severity of injury also matters: the risk of post-TBI seizures is high in the first year in patients with severe TBI and significantly increases up to 10 years for moderate TBI [20]. In this study, the median follow-up period was only 59 months; hence, we were unable to correlate seizure occurrence and its temporal evolution over time in relation to the severity of initial TBI. However, we noted additional demographic factors such as the finding that those patients who were employed at the time of TBI showed an increased tendency of developing post-TBI seizures (Table 1; *p*-value = 0.097). This finding could be explained by the possibility that the younger patients in this cohort might have been involved in more violent injuries as a result of road traffic accidents in contrast with older adults who might have presented with relatively minor and less violent injuries (*p*-value = 0.132, which did not reach significance in this study).

Another peculiar finding that may deserve more attention in the future is that, in this cohort, liver dysfunction at the time of admission was associated with long term post-TBI seizure occurrence (Table 2). The reason for this is unclear, though a previous report seems to suggest a correlation f liver cirrhosis with the occurrence of TBI itself [21].

Regarding antiepileptic drugs, phenytoin and levetiracetam are the most common prophylactic agents used for post-TBI seizures, with phenytoin being recommended by the American Academy of Neurology for the prevention of early post-traumatic seizures [1,22,23,24]. In our study, the most common antiepileptic agent used was indeed levetiracetam, followed by valproic acid and phenytoin. Neuroplasticity and its role in brain injury recovery has been well studied. However, there are very few neuromodulators that are currently available for use in TBI patients: amantadine, methylphenidate, modafinil, levodopa, and citalopram have been shown to benefit cognitive improvement and recovery following brain injury [25,26]. Bromocriptine was shown to improve outcomes following TBI, possibly by alleviating autonomic dysfunction [17]. In the patients analyzed in our study, the most common neuromodulator used was levodopa, followed by bromocriptine, which is indicated to manage autonomic instability following TBI. Surprisingly, we found a statistically significant association between the use of these agents and seizure occurrence (39.4% vs. 12%; *p*-value < 0.001). One possible explanation is that patients who were prescribed levodopa or bromocriptine were more likely to have more severe brain injuries. Hence, the relationship between seizure occurrence in these patients may have been indicative of the severity of brain injury rather than the use of neurostimulants, though this latter possibility cannot be completely ruled out. Of interest, we previously reported a possible correlation of levodopa use in stroke patients with the occurrence of post-stroke seizures [27]. As such, the data presented here need to be carefully interpreted with respect to the role of neurostimulant use in post-TBI vs. post-stroke seizures, and future prospective trials are required to validate these findings.

We did not find any protective effect of lipid-lowering agents in post TBI seizures, although the protective effect was found in another study of post-stroke seizures [27]. This could have been due to the small number of patients analyzed in the current TBI cohort.

### 4.3. Future Work

In addition to neurostimulants and the use of statins, future studies can investigate the role of various biomarkers to predict the occurrence of post-TBI seizures and to potentially guide future therapeutics. Elevated levels of IL-6 are observed in TBI patients [28] and are considered to play a significant role in those with temporal lobe epilepsy [29] and pediatric epilepsy [30]. The CSF–serum IL-1β ratio was shown to be elevated in TBI patients who are prone to developing epilepsy [31]. Treatment with IL-1 β receptor antagonist reduced seizures in a mouse model [30]. Other studies suggested the use of lipopolysaccharide to prevent the acceleration of kindling epileptogenesis as a result of TBI [32]. Another possible area of exploration is with the manipulation of microglial cells, whose activation is commonly seen following TBI and seizure occurrence [33]. Microglia play a significant role in seizure occurrence, and their inhibition with minocycline has been shown to reduce post TBI seizures and cognitive deficits; this relationship warrants further evaluation [6,34,35,36,37]. Finally, from the clinical perspective, an effort to limit the use of levodopa or bromocriptine in TBI patients may be tested for its possible role in mitigating the risk of post-traumatic seizures.

## 5. Conclusions

In the current study of 141 patients from the multiracial nation of Singapore, we report a post-traumatic seizure rate following TBI of 24.4%. Patients who are of younger age and patients who received neurostimulants are at a higher risk of seizures following TBI. Adequate information should be provided to the patients at risk regarding long-term seizure occurrence. This is important from a medicolegal point of view and from the viewpoint of compensation claims. The role of neurostimulants and their relationship with seizure occurrence needs to be investigated in further studies.

## Figures and Tables

**Table 1 ijerph-20-02301-t001:** Differing demographical and medical characteristics of patients with traumatic brain injury vis à vis seizure occurrence.

Parameter	Seizure (n = 33)	No Seizure (n = 108)	*p*-Value
Age, mean (SD)	53.0 (19.5)	63.6 (19.1)	**0.008**
Sex, n (%)			0.363
Female	6 (18.2)	28 (25.1)	
Male	27 (81.8)	80 (74.9)	
Race, n (%)			**0.035**
Chinese	18 (54.5)	74 (68.5)	
Malay	9 (27.3)	30 (27.8)	
Indian	4 (12.1)	2 (1.9)	
Other	2 (6.1)	2 (1.9)	
Occupation, n (%)			0.097
Working	21 (63.6)	46 (42.6)	
Retired	11 (33.3)	59 (54.6)	
Mode of injury, n (%)			0.132
Fall	22 (68.8)	65 (63.1)	
Road traffic accident	8 (25.0)	37 (35.9)	
Industrial accident	2 (6.3)	1 (1.0)	
History of diabetes, n (%)			0.816
Yes	7 (21.2)	25 (23.1)	
No	26 (78.8)	83 (76.9)	
History of hypertension, n (%)			0.675
Yes	13 (39.4)	47 (43.5)	
No	20 (60.6)	61 (56.5)	
History of hyperlipidemia, n (%)			0.673
Yes	12 (36.4)	35 (32.4)	
No	21 (63.6)	73 (67.6)	
History of chronic kidney disease, n (%)			0.557
Yes	2 (6.1)	4 (3.7)	
No	31 (93.9)	104 (96.3)	
Atrial fibrillation, n (%)			0.564
Yes	2 (6.1)	10 (9.3)	
No	31 (93.9)	98 (90.7)	
Ischemic heart disease, n (%)			0.384
Yes	7 (21.2)	16 (14.8)	
No	26 (78.8	92 (85.2)	
Anemia, n (%)			0.891
Yes	3 (9.1)	9 (8.3)	
No	30 (91.7)	99 (91.7)	
Past history of stroke, n (%)			0.795
Yes	4 (12.1)	15 (13.9)	
No	29 (87.9)	93 (86.1)	
Presence of liver dysfunction, n (%)			**0.01**
Yes	2 (6.1)	0 (0)	
No	31 (93.9)	108 (100)	
Subdural hemorrhage, n (%)			0.386
Yes	18 (54.5)	68 (63.0)	
No	15 (45.5)	40 (37.0)	
Subarachnoid hemorrhage, n (%)			0.882
Yes	13 (39.4)	41 (38.0)	
No	20 (60.6)	67 (62.0)	
Cerebral contusion, n (%)			**0.05**
Yes	22 (66.7)	51 (47.2)	
No	11 (33.3)	57 (52.8)	
Intraventricular hemorrhage, n (%)			0.263
Yes	8 (24.2)	17 (15.7)	
No	25 (75.8)	91 (84.3)	
Extradural hemorrhage, n (%)			0.31
Yes	7 (21.2)	15 (13.9)	
No	26 (78.8)	93 (86.1)	
Neurosurgical intervention, n (%)			**<0.001**
No intervention	12 (37.5)	56 (51.9)	
EVD-ICP	6 (18.8)	13 (12.0)	
Burr-hole	1 (3.1)	24 (22.2)	
Craniotomy	1 (3.1)	7 (6.5)	
Craniectomy	9 (28.1)	8 (7.4)	
V-P shunt	3 (9.4)	0 (0)	
GCS at admission, mean (SD)	9.5 (4.1)	11.9 (3.0)	**0.003**
Hospital length of stay at index admission, day, mean (SD)	53.1 (36.7)	36.3 (30.1)	**0.021**
Use of neurostimulants, n (%)			**<0.001**
Yes	13 (39.4)	13 (12.0)	
No	20 (60.6)	95 (88.0)	
Use of Statins, n (%)			0.207
Yes	10 (30.3)	46 (42.6)	
No	23 (69.7)	62 (57.4)	
Mortality, n (%)			0.873
Death	9 (27.3)	31 (28.7)	
Alive	24 (72.7)	77 (71.3)	
Follow-up period, days, median (25–75%)	2242 (951–2894)	1435 (845–2543)	0.082
Hb, mean (SD)	13.3 (2.1)	13.2 (2.2)	0.919
WBC, mean (SD)	13.4 (67)	11.5 (4.3)	0.141
Platelet, mean (SD)	276.6 (104.5)	272.3 (90.2)	0.833
MCV, mean (SD)	86.2 (8.2)	88.4 (6.8)	0.179
MCH, mean (SD)	28.6 (3.3)	29.4 (3.9)	0.289
Urea, mean (SD)	4.7 (2.6)	5.6 (2.8)	0.101
Na, mean (SD)	137 (4.7)	137.4 (4.3)	0.687
K, mean (SD)	3.9 (0.5)	4.0 (0.5)	0.299
Bicarb, mean (SD)	21.9 (3.6)	22.8 (3.1)	0.207
Creatinine, mean (SD)	122.8 (186.4)	98.9 (50.3)	0.472
eGFR, mean (SD)	57.5 (12.4)	55.8 (10.3)	0.484

Results: Occurrence of seizures after traumatic brain injury was significantly associated with younger age, being Indian, presence of liver dysfunction, neurosurgical interventions, consciousness level at admission, and use of neurostimulant medications. Bold face indicates *p*-values < 0.05.

**Table 2 ijerph-20-02301-t002:** Multivariate analysis.

Parameter	B	S.E.	Wald	df	*p*-Value	OR	95% CI for OR	
							Lower	Upper
Age	−0.046	0.013	11.422	1	0.001	0.956	0.931	0.981
Neurostimulants	2.389	0.584	16.745	1	0.0001	10.899	3.472	34.220
Liver dysfunction	24.165	28073.795	0.000	1	0.999		0.000	
Constant	0.359	0.699	0.263	1	0.608	1.431		

## Data Availability

Data from this research study are available upon request.

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
