# Peer review of "Young Age, Liver Dysfunction, and Neurostimulant Use as Independent Risk Factors for Post-Traumatic Seizures: A Multiracial Single-Center Experience"

_ijerph, 2023, doi:10.3390/ijerph20032301_

Round 1
Reviewer 1 Report
Interesting article. My comments: Abstract: line 17: please define GCS abbreviation. Introduction : Please provide a clear research hypothesis. methods : please define what parameters and in what values ​​indicated the correct and the wrong liver function. What ethnicity (races) were evaluated?Author Response
Thank you, Reviewer 1, for your insightful comments that help improve our Article.
Abstract: line 17: please define GCS abbreviation.
Thank you for this important reminder. Since GCS stands for Glasgow Coma Scale, we have now revised the Abstract (line 16-18) to read:
Young age, presence of cerebral contusion, Indian race, low Glasgow Coma Scale (GCS) scores on admission and use of neurostimulant medications were associated with increased risk of seizures.
Introduction : Please provide a clear research hypothesis.
Page 2 Line 48: we have added the following research hypothesis:
We sought to characterize potentially modifiable risk factors that may predispose patients to developing post-traumatic seziures. We hypothesized that in addition to the commonly known risk factors such as age, there may be other factors such as drug use that are hitherto unknown, which may predict the risk and if modified may reduce the chances of patients developing post-traumatic seizures.
Methods : please define what parameters and in what values ​​indicated the correct and the wrong liver function. What ethnicity (races) were evaluated?
Thank you, Reviewer for these constructive comments. We have added the following to the Methods section Page 3 Line 100:
Liver function tests were defined as abnormal if the patient’s Alanine Aminotransferase (ALT) and Aspartate Aminotransferase (AST) levels were higher than two times the Upper Limit of Normal (ULN).
We have also added this sentence on Page 2 Line 60:
All patients, regardless of racial background (Chinese, Malay, Indian, or others) admitted with a diagnosis of TBI have their neurological status and initial GCS assessed.

Reviewer 2 Report
Dear Authors; I found this work an interesting investigation to determine potentially modifiable risk factors that are predictive of post-traumatic brain injury seizures in relation to severity of initial injury, neurosurgical interventions, neurostimulant useand comorbidities. Prior to processing it further, it needs some extra work. Regards. P.S.
[1] Writing:
[1-1] Missing Abbreviations: Add list of used abbreviations in the text right before Reference section for readers referral.
[1-2] Reference Format: Make sure these are in MDPI format. For example, for papers years are in bold font, etc.
[1-3] Missing outline of the paper paragraph: Add this paragraph in line 51.
[1-4] Missing subsection numbers: Add these . Example: "2.1. Patient Selection".
[1-5] Substandard Result Section: Break it down into two subsections for clarity of the read: "3.1. Descriptive Analysis". "3.2. Multivariate Regression Analysis".
[1-6] Discussion section: Break it down into several subsections for clarity of the read: "4.1. Summary & Contributions", "4.2. Strength & Limitations", "4.3. Future Work".
[1-7] In section "4.3. Future Work" add a paragraph and propose some future research directions.
[2] Statistical:
[2-1] How did you select covariates for MVL logistic regression ? Add it to the text.
[2-2] Did you check interacctions in your MVL logistic regression ? Report it in the text.
[2-3] Missing Story Figure: Add a Figure in line 132 to deliver the main message to the reader in that figure wit the following details(We want to see the trajectory of experimental/control status over age groups):
X-axis: age
Y-axis: log odds/odds of outcome (95%CI) per group
See the following paper (last figure for some ideas):
Link: https://www.mdpi.com/1660-4601/16/8/1347/htm
Author Response
Dear Authors; I found this work an interesting investigation to determine potentially modifiable risk factors that are predictive of post-traumatic brain injury seizures in relation to severity of initial injury, neurosurgical interventions, neurostimulant use and comorbidities. Prior to processing it further, it needs some extra work. Regards. P.S.
[1] Writing:
[1-1] Missing Abbreviations: Add list of used abbreviations in the text right before Reference section for readers referral.
RESPONSE: We have added the list of used abbreviations, with many thanks to Reviewer 2! The list is as follows:
GCS = Glasgow Coma Scale
TBI = Traumatic Brain Injury
CSF = Cerebrospinal Fluid
EEG = Electroencephalogram
PTA = Post-Traumatic Amnesia
ALT = Alanine Aminotransferase
AST = Aspartate Aminotransferase
CT = Computed Tomography
ICH = Intracerebral Haemorrhage
SAH = Subarachnoid Haemorrhage
SDH = Subdural Haemorrhage
IVH = Intraventricular Haemorrhage
CI = Confidence Interval
SD = Standard Deviation
ICP = Intracranial Pressure
EVD = Extraventricular Drainage
Hb = Haemoglobin
WBC = White Blood Cell
MCV = Mean Corpuscular Volume
MCH = Mean Corpuscular Haemoglobin
eGFR = Estimated Glomerular Filtration Rate
[1-2] Reference Format: Make sure these are in MDPI format. For example, for papers years are in bold font, etc.
RESPONSE: We have reformatted every reference to be in MDPI format.
[1-3] Missing outline of the paper paragraph: Add this paragraph in line 51.
RESPONSE: We have added the following lines:
We sought to characterize potentially modifiable risk factors that may predispose patients to developing post-traumatic seziures. We hypothesized that in addition to the commonly known risk factors such as age, there may be other factors such as drug use that are hitherto unknown, which may predict the risk and if modified may reduce the chances of patients developing post-traumatic seizures. In this study, we analyzed 141 patients with TBI, and confirmed as per previous publications that the non-modifiable risk factors of younger age, poor initial GCS scores, and the existence of cerebral contusion on brain imaging are indeed risk factors predictive of post-TBI seizures. In addition, we identify novel potentially modifiable risk factors such as liver dysfunction, active employment, and the use of neurostimulants as potential risk factors which deserve further study.
[1-4] Missing subsection numbers: Add these . Example: "2.1. Patient Selection".
RESPONSE: Subsections added.
[1-5] Substandard Result Section: Break it down into two subsections for clarity of the read: "3.1. Descriptive Analysis". "3.2. Multivariate Regression Analysis".
RESPONSE: Subsections added.
[1-6] Discussion section: Break it down into several subsections for clarity of the read: "4.1. Summary & Contributions", "4.2. Strength & Limitations", "4.3. Future Work".
RESPONSE: Subsections added as suggested by Reviewer 2.
[1-7] In section "4.3. Future Work" add a paragraph and propose some future research directions.
RESPONSE: Future Work was added with some future research directions included.
[2] Statistical:
[2-1] How did you select covariates for MVL logistic regression ? Add it to the text.
RESPONSE:This study aimed to identify potential risk factors that may be associated with post-traumatic seizures. Variables that showed significant association with seizures in univariate logistics regression were selected for MVL regression (line 113).
[2-2] Did you check interacctions in your MVL logistic regression ? Report it in the text.
RESPONSE: Thank you reviewer for this comment. We did not check for interaction in MVL logistics regression. The reason for this is we did not find any variable with a large enough effect on the outcome variables when variables were included in MVL. On top of that, the direction of effect is consistent for both univariate and multivariate analysis.
[2-3] Missing Story Figure: Add a Figure in line 132 to deliver the main message to the reader in that figure wit the following details(We want to see the trajectory of experimental/control status over age groups):
X-axis: age
Y-axis: log odds/odds of outcome (95%CI) per group
See the following paper (last figure for some ideas):
Link: https://www.mdpi.com/1660-4601/16/8/1347/htm
RESPONSE: Thank you so much, Reviewer for this suggestion. As this is a study that aims to identify potential risk factors that associated with post-traumatic seizures, there is no experimental/control group in this study. Hence, we could not by design show the trajectory of experimental/control status over age groups.

Reviewer 3 Report
The authors studied 33 pts suffering from posttraumatic seizures out of 141 pts who were rehabilitated following severe TBI. 13 of the 33 pts were on various neuro stimulants, and 20 did not get any drugs belonging to „neuro stimulants”. The type of seizures and pathology explaining the development of the seizures (focal, diffuse, etc...…) are extensive. Imaging modality is not uniform. The lengths of neuro-stimulant treatment are unknown. Indication for administering neuro stimulants is also unknown both in the seizure and the seizure-free groups.
Based on multivariate analysis, these authors have concluded that young age, presence of cerebral contusion, Indian race, low GCS on admission and use of neuro stimulant medications were associated with an increased risk of seizures. This finding led the authors to conclude that due to the increased risk of seizures, younger TBI patients, patients with low GCS on admission, cerebral contusions on brain imaging, and those who received neuro stimulants or neurosurgical interventions should be monitored for post-TBI seizures.
Unfortunately, the data presented do not provide statistically valid scientific proof that this selected group of TBI pts. needs to be monitored for posttraumatic seizures, as this is mandatory for all pts suffering from severe TBI.
Based on an uncertain retrospective database with a small number of cases, it cannot be told whether there is a connection between the neurostimulation treatment and the development of post-TBI seizures! It is an exciting proposition; therefore, a targeted study with a reasonable statistical hypothesis should be contemplated on a multicentric level.
Author Response
Reviewer 3, thank you very much for your insightful comments.
The authors studied 33 pts suffering from posttraumatic seizures out of 141 pts who were rehabilitated following severe TBI. 13 of the 33 pts were on various neuro stimulants, and 20 did not get any drugs belonging to „neuro stimulants”. The type of seizures and pathology explaining the development of the seizures (focal, diffuse, etc...…) are extensive. Imaging modality is not uniform. The lengths of neuro-stimulant treatment are unknown. Indication for administering neuro stimulants is also unknown both in the seizure and the seizure-free groups.
RESPONSE:
- The type of seizures and pathology explaining the development of the seizures (focal, diffuse, etc...…) are extensive. à We agree with Reviewer 3. It is imperative in retrospective studies such as this one, that we do not limit our analysis to just one type or pathology of seizures.
- Imaging modality is not uniform. à We outlined on page 2 line 65 that the imaging modality in this study was CT brain. We have now revised the line to read:
All patients, regardless of racial background (Chinese, Malay, Indian, or others) admitted with a diagnosis of TBI have their neurological status and initial GCS assessed. All patients received uniformly urgent CT brain scan, baseline full blood count, biochemistry, liver function tests and coagulation profile at baseline presentation. Based on the initial and subsequent neurological status and neuroimaging by CT of the brain, appropriate referrals to the neurosurgical team to perform any surgical intervention were made.
- The indication for the neurostimulant medications (levodopa, bromocriptine) was outlined in our Discussion section. We refer Reviewer 3 to Page 7, Line 200-207:
Neuroplasticity and its role in brain injury recovery has been well studied. However, there are very few neuromodulators which are currently available for use in TBI patients. Amantadine, methylphenidate, modafinil, levodopa and citalopram have been shown to benefit cognitive improvement and recovery following brain injury 25,26. Bromocriptine has been shown to improve outcomes following TBI, possibly by alleviating autonomic dysfunction 17. In the patients analyzed in our study, the most common neuromodulator used was levodopa, followed by bromocriptine which was indicated to manage autonomic instability following TBI.
REVIEWER COMMENT:
Based on multivariate analysis, these authors have concluded that young age, presence of cerebral contusion, Indian race, low GCS on admission and use of neuro stimulant medications were associated with an increased risk of seizures. This finding led the authors to conclude that due to the increased risk of seizures, younger TBI patients, patients with low GCS on admission, cerebral contusions on brain imaging, and those who received neuro stimulants or neurosurgical interventions should be monitored for post-TBI seizures.
Unfortunately, the data presented do not provide statistically valid scientific proof that this selected group of TBI pts. needs to be monitored for posttraumatic seizures, as this is mandatory for all pts suffering from severe TBI.
RESPONSE: Indeed, the monitoring for posttraumatic seizures is the gold standard for all patients suffering from severe TBI. Nevertheless, though it is generally accepted that older patients and those with more severe injuries are at a higher risk for developing posttraumatic seizures, other as of yet unidentified factors such as the use of levodopa or bromocriptine which was identified in this study is a novel and importantly, possibly modifiable risk factor that may influence future clinical practice. We concur otherwise with Reviewer 3 as per our Discussion section in which we stated:
One possible explanation is that patients who were prescribed levodopa or bromocriptine were more likely to be patients with more severe brain injuries, hence the relationship between seizure occurrence in these patients may be indicative of the severity of brain injury rather than the use of neurostimulants per se, though this latter possibility cannot be completely ruled out. Of interest, we previously reported the possible correlation of levodopa use in stroke patients with the occurrence of post-stroke seizures27. As such, the data presented here needs to be interpreted carefully with respect to the role of neurostimulant use in post-TBI vs post-stroke seizures and future prospective trials are required to validate these findings.
Based on an uncertain retrospective database with a small number of cases, it cannot be told whether there is a connection between the neurostimulation treatment and the development of post-TBI seizures! It is an exciting proposition; therefore, a targeted study with a reasonable statistical hypothesis should be contemplated on a multicentric level.
RESPONSE: We completely understand Reviewer 3's reservations about the conclusions drawn. We would like to point out that in this single center retrospective analysis, we have found through an analysis of 141 Traumatic Brain Injury patients that the use of neurostimulants (such as bromocriptine and levodopa) conferred an increased risk to developing post-TBI seizures with a statistically significant Odds Ratio (OR=10.9,95% CI:3.5-34.2, p=<0.001); hence, although it might not be obvious or predictable to date, this finding represents the beginning of future studies that may look into this in greater detail.
We have also revised the Introduction section to reflect our scientific hypothesis:
Page 2 line 48-58:
We sought to characterize potentially modifiable risk factors that may predispose patients to developing post-traumatic seziures. We hypothesized that in addition to the commonly known risk factors such as age, there may be other factors such as drug use that are hitherto unknown, which may predict the risk and if modified may reduce the chances of patients developing post-traumatic seizures. In this study, we analyzed 141 patients with TBI, and confirmed as per previous publications that the non-modifiable risk factors of younger age, poor initial GCS scores, and the existence of cerebral contusion on brain imaging are indeed risk factors predictive of post-TBI seizures. In addition, we identify novel potentially modifiable risk factors such as liver dysfunction, active employment, and the use of neurostimulants as potential risk factors which deserve further study.

Round 2
Reviewer 2 Report
Dear Authors, my main concerns were addressed satisfactorily. Regards.
Reviewer 3 Report
I still believe that due to an uncertain retrospective database with a small number of cases, it cannot be told whether there is a real connection between the neurostimulation treatment and the development of post-TBI seizures! However, it is an exciting proposition; therefore, a targeted study with a reasonable statistical hypothesis should be contemplated on a multicentric level.